# Chain-of-Retrieval Augmented Generation

**Liang Wang**[†][*]  **Haonan Chen**[‡]  **Nan Yang**[†]  **Xiaolong Huang**[†]  **Zhicheng Dou**[‡]  **Furu Wei**[†]

[†]Microsoft Research   [‡]Renmin University of China

https://aka.ms/GeneralAI

## Abstract

This paper introduces an approach for training o1-like RAG models that retrieve and reason over relevant information step by step before generating the final answer. Conventional RAG methods usually perform a single retrieval step before the generation process, which limits their effectiveness in addressing complex queries due to imperfect retrieval results. In contrast, our proposed method, **CoRAG** (**C**hain-**o**f-**R**etrieval **A**ugmented **G**eneration), allows the model to dynamically reformulate the query based on the evolving state. To train CoRAG effectively, we utilize rejection sampling to automatically generate intermediate retrieval chains, thereby augmenting existing RAG datasets that only provide the correct final answer. At test time, we propose various decoding strategies to scale the model's test-time compute by controlling the length and number of sampled retrieval chains. Experimental results across multiple benchmarks validate the efficacy of CoRAG, particularly in multi-hop question answering tasks, where we observe more than 10 points improvement in EM score compared to strong baselines. On the KILT benchmark, CoRAG establishes a new state-of-the-art performance across a diverse range of knowledge-intensive tasks. Furthermore, we offer comprehensive analyses to understand the scaling behavior of CoRAG, laying the groundwork for future research aimed at developing factual and grounded foundation models.

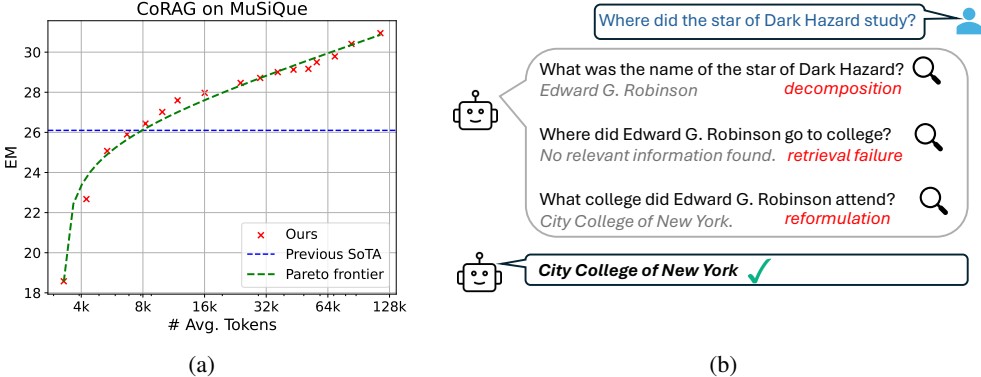

Figure 1: (a) Test-time scaling behavior of CoRAG. Increased token budget leads to consistent performance improvements. (b) An example of CoRAG on the MuSiQue dataset. It learns to decompose the complex query and conduct query reformulation when encountering a retrieval failure.

---

[*]Correspondence to wangliang@microsoft.com

39th Conference on Neural Information Processing Systems (NeurIPS 2025).

# 1 Introduction

Retrieval-augmented generation (RAG) [20] is one of the core techniques in enterprise applications, necessitating the integration of large foundation models with proprietary data sources to produce responses that are both grounded and factual. Conventionally, foundation models are trained on large-scale datasets comprising trillions of tokens and remain frozen post-deployment. Nonetheless, these models frequently struggle to memorize long-tail factual knowledge or may hallucinate false claims, resulting in unreliable responses in real-world scenarios. RAG mitigates this challenge by augmenting the generation process with retrieved information, thereby improving the trustworthiness of model-generated content and facilitating the incorporation of up-to-date information.

Contemporary RAG systems typically employ a sequential pipeline of retrieval and generation, wherein the retrieved information serves as additional input to the generative model. The effectiveness of RAG systems predominantly relies on the quality of the retrieved information. Retrieval models are engineered for efficiency to ensure scalability to large corpora. For instance, dense retrievers [18, 35] commonly utilize a bi-encoder architecture to compress documents and queries into fixed-size vector representations. This architectural choice permits the use of fast approximate nearest neighbor search algorithms but simultaneously constrains the expressive capacity of retrieval models to handle complex queries. Furthermore, in multi-hop reasoning tasks, it is often unclear what information should be retrieved initially; decisions must be made based on the progressively evolving state of the reasoning process.

To break the bottleneck of retrieval quality, we propose a framework that dynamically retrieves relevant information and plans subsequent retrieval steps based on the current state. By adjusting the number of retrieval steps at test time, our model can explore various aspects of the query and experiment with different query rewriting strategies when the retriever does not yield useful information. This paradigm mirrors the human problem solving process, where we iteratively seek information to address complex questions. An example is illustrated in Figure 1.

Rather than solely relying on the model's in-context learning capability [42] or distillation from proprietary models [1], we advocate for explicitly training language models to retrieve step by step. To this end, we utilize rejection sampling [43, 5] to augment existing RAG datasets with intermediate retrieval chains. Open-source language models are then fine-tuned on these augmented datasets using standard next-token prediction objectives. To examine the scaling behavior of our model, we propose various test-time decoding strategies, including greedy decoding, best-of-$N$ sampling, and tree search. Diverse decoding strategies and hyperparameter configurations can be employed to control test-time token consumption and the frequency of retriever calls.

Our empirical evaluation demonstrates that CoRAG substantially surpasses strong baselines in QA tasks that require multi-hop reasoning, where retrievers frequently struggle to recall all necessary information in a single retrieval step. Across diverse decoding strategies, the Pareto frontier approximately adheres to a log-linear relationship between total token consumption and model performance, although the coefficients differ across datasets.

On the KILT benchmark [27], which encompasses a more diverse array of tasks, new state-of-the-art scores are achieves on the *hidden test set* for nearly all tasks. Additionally, we uncover that CoRAG exhibits varied scaling behaviors across different task types. For datasets such as NQ [19], where state-of-the-art retrievers already achieve high recall, the benefits of test-time scaling are often marginal. This suggests the potential for dynamically allocating test-time compute based on the complexity of the query and the quality of the retriever. Upon further analysis, we find that CoRAG can effectively decompose complex queries and perform flexible query reformulation to improve the quality of the generated responses. It also shows robustness against retrievers of varying quality. We posit that CoRAG represents a promising avenue for future research in the RAG domain, with the potential to mitigate hallucination in model-generated content. Our code, data and trained models are available at https://github.com/microsoft/LMOps/tree/main/corag.

# 2 Related Work

**Retrieval-Augmented Generation (RAG)** integrates information retrieval techniques with generative models to enhance the quality and factual accuracy of generated content [20, 21]. By equipping LLMs with the ability to browse the web [26], RAG systems can access real-time data, thereby

providing responses that are both up-to-date and grounded. The relevance and quality of the retrieved information are pivotal for the efficacy of RAG systems. A substantial body of recent research has concentrated on developing better general-purpose text embeddings [18, 35]. Nevertheless, text embeddings frequently face limitations in addressing complex queries due to their reliance on fixed-size vector representations for efficiency purposes.

To mitigate this constraint, contemporary research has extended the conventional paradigm of a single retrieval step followed by generation, advancing to multi-step iterative retrieval and generation [6]. FLARE [13] prompts an LLM to actively determine when and what to retrieve during the generation process. ITER-RETGEN [30] proposes to interleave retrieval-augmented generation with generation-augmented retrieval, demonstrating enhancements in multi-hop QA tasks. Similarly, IRCoT [33] employs a chain-of-thought methodology, which recursively refines the reasoning thought for subsequent retrieval steps. Self-RAG [1] empowers LLMs to adaptively retrieve, generate, and critique through self-reflection, thus improving factual accuracy and citation precision in open-domain QA and long-form generation tasks. Auto-RAG [41] utilizes heuristic rules and exact answer matching to construct intermediate retrieval steps, yet its performance remains significantly below that of state-of-the-art models. AQA [3] learns to reformulate questions using reinforcement learning but only focuses on single-hop QA tasks. In this study, rather than exclusively on few-shot prompting or distillation from proprietary models, we propose a novel approach to explicitly train LLMs to iteratively retrieve and reason over relevant information.

**Scaling Test-time Compute** Instead of prompting LLMs to directly generate the final answer, Chain-of-Thought (CoT) [36] demonstrates that letting the model to think step by step can drastically improve the performance on mathematical reasoning tasks. Tree-of-Thought (ToT) [40] extends the idea of CoT by adopting a tree structure, allowing the model to explore the search space more comprehensively. To further enhance the reasoning capabilities of LLMs, STaR [43] proposes to leverage bootstrapping techniques to generate intermediate states for training. OpenAI o1 [12] conducts large-scale reinforcement learning and exhibits promising test-time scaling behaviors on advanced reasoning datasets, but the technical details are not publicly available. A drawback of these methods is the increased token consumption, which consequently increases the response latency.

In the realm of RAG, test-time compute can be increased by retrieving more documents or performing additional retrieval steps. LongRAG [14] posits that RAG performance can be enhanced by integrating long-context LLMs with more retrieved documents. In contrast, IterDRAG [42] empirically examines the test-time scaling law through few-shot prompting and iterative retrieval for up to 5M tokens. Search-o1 [22] combines the open-source QwQ model [37] with active search from Bing, achieving competitive results on knowledge-intensive tasks. Concurrent works such as Search-R1 [15] train LLMs to use retrieval as a tool via reinforcement learning. Our work extends the study of test-time scaling in RAG to a targeted fine-tuning paradigm under diverse decoding strategies.

## 3 Methodology

The CoRAG framework is illustrated in Figure 2. The "Current State" denotes the input context and instructions provided to the LLM, while the "Next Action" refers to the LLM output responding to the given instruction. In this section, we describe the key components of CoRAG, including retrieval chain generation through rejection sampling, model training with augmented datasets, and strategies for scaling test-time compute.

### 3.1 Retrieval Chain Generation

Most RAG datasets only come with a query $Q$ and the corresponding final answer $A$, without providing intermediate retrieval steps. We propose an automated method for generating retrieval chains through rejection sampling. Each sampled chain consists of a sequence of sub-queries $Q_{1:L} = \{Q_1, Q_2, \ldots, Q_L\}$ and the corresponding sub-answers $A_{1:L}$, where $L$ is a predetermined maximum chain length. The sub-query $Q_i = \text{LLM}(Q_{<i}, A_{<i}, Q)$ is generated by sampling an LLM based on the query $Q$ and the preceding sub-queries and sub-answers. To generate the sub-answer $A_i$, we first retrieve the top-$k$ most relevant documents $D_{1:k}^{(i)}$ using a text retriever with $Q_i$ as the search

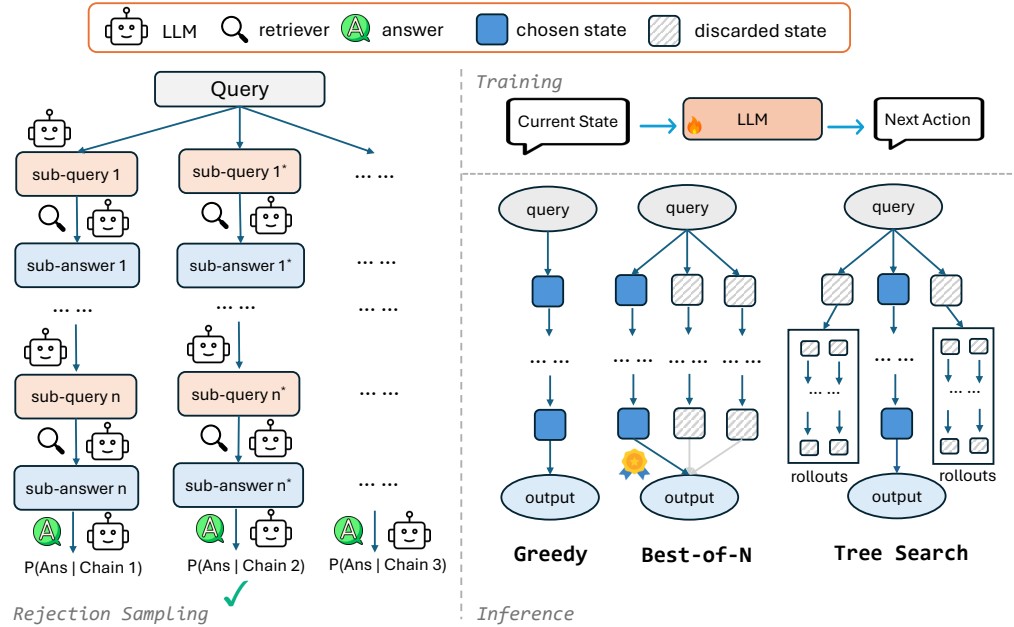

Figure 2: Overview of CoRAG. Rejection sampling is utilized to augment QA-only datasets with retrieval chains. Each chain starts with the original query, followed by a sequence of sub-queries and sub-answers. An open-source LLM is then fine-tuned to predict the next action based on the current state. During inference, multiple decoding strategies are available to control the test-time compute.

query, and subsequently prompt an LLM to yield the answer $A_i = \text{LLM}(Q_i, D_{1:k}^{(i)})$. This procedure is iterated until the chain reaches the maximum length $L$ or $A_i$ matches the correct answer $A$.

To assess the quality of a retrieval chain, we calculate the log-likelihood of the correct answer $\log \text{P}(A|Q, Q_{1:L}, A_{1:L})$ conditioned on the chain information. The retrieval chain with the highest log-likelihood score is selected to augment the original QA-only dataset.

## 3.2 Training

Each training instance in the augmented dataset is represented as a tuple $(Q, A, Q_{1:L}, A_{1:L})$, accompanied by the corresponding top-$k$ retrieved documents for the query $Q$ and each sub-query. We fine-tune an LLM on the augmented dataset using the standard next-token prediction objective within a unified multi-task learning framework.

The model is simultaneously trained on three tasks: next sub-query prediction, sub-answer prediction, and final answer prediction. We employ the same prompt templates as utilized in the retrieval chain generation process, with the exception that we also incorporate the top retrieved documents $D_{1:k}$ for the original query $Q$ as input for the final answer prediction task.

$$L_{\text{sub\_query}} = -\log \text{P}(Q_i|Q, Q_{<i}, A_{<i}), i \in [1, L]$$

$$L_{\text{sub\_answer}} = -\log \text{P}(A_i|Q_i, D_{1:k}^{(i)}), i \in [1, L]$$

$$L_{\text{final\_answer}} = -\log \text{P}(A|Q, Q_{1:L}, A_{1:L}, D_{1:k})$$

The cross-entropy loss is computed only for the target output tokens. As we reuse the prompt templates for both data generation and model training, a fine-tuned model can be utilized for the next round of rejection sampling in an iterative manner.

## 3.3 Test-time Scaling

Given a trained CoRAG model, we propose several decoding strategies to control the trade-off between model performance and test-time compute. The test-time compute is measured by the

total number of token consumptions, excluding the retrieval costs. Unlike previous approaches that consider only prompt tokens [42] or generated tokens [12], we account for both. To simplify further discussion, the prompt tokens are treated equally as the generated tokens, despite prompt tokens typically being less expensive due to prefix caching and computation parallelism of the prefilling stage.

**Greedy Decoding** This strategy utilizes greedy decoding to generate $L$ sub-queries and their corresponding sub-answers sequentially. The final answer is generated using the same prompt template as employed during the training phase.

**Best-of-$N$ Sampling** This method involves sampling $N$ retrieval chains with a temperature 0.7, subsequently selecting the best chain to generate the final answer. As the ground truth answer is not available at test time, we instead calculate the conditional log-likelihood of *"No relevant information found"* as a penalty score for each chain. The retrieval chain with the lowest penalty score is chosen.

**Tree Search** We implement a breadth-first search (BFS) variant with retrieval chain rollouts. At each step, the current state is expanded by sampling several sub-queries. For each expanded state, we perform multiple rollouts, and then compute the average penalty score of these rollouts. The state with the lowest average penalty score is retained for further expansion.

To control the test-time compute, the maximum length of the retrieval chain $L$ can be adjusted across all decoding strategies. For best-of-$N$ sampling, the number of sampled chains $N$ offers an alternative option to scale the test-time compute. In tree search, the number of rollouts and expansion size are two additional hyperparameters.

# 4 Experiments

## 4.1 Setup

**Data and Evaluation** We evaluate CoRAG utilizing two sets of benchmarks: (1) a collection of multi-hop QA datasets, including 2WikiMultihopQA [8], HotpotQA [39], Bamboogle [28], and MuSiQue [32]; (2) the KILT benchmark [27], which encompasses a broad spectrum of knowledge-intensive tasks. The multi-hop QA datasets serve to evaluate the model's capacity to perform multi-hop reasoning, whereas the KILT benchmark assesses the framework's ability to generalize across more diverse tasks. For each training dataset, we prompt the open-source *Llama-3.1-8B-Instruct* model to perform rejection sampling, unless specified otherwise. We utilize E5-large [34] as the text retriever for intermediate retrieval steps. The retrieval corpus is the English Wikipedia provided by KILT, comprising approximately 36 million passages [25]. The selected retrieval chains are employed to augment the original QA-only datasets for subsequent model training.

Regarding evaluation metrics, we report the exact match (EM) and F1 scores [29] for the multi-hop QA datasets. For the KILT benchmark, we submit the model's predictions to the official evaluation server and report the downstream metrics on the *hidden test set*. To adhere to the leaderboard submission policy, we report *public validation set* results when conducting ablation studies on the KILT benchmark.

Note that while HotpotQA and MuSiQue maintain public leaderboards, these adopt either a simplified reading comprehension setting or an abstract-only retrieval configuration. Consequently, the leaderboard results are not directly comparable to our open-domain QA evaluation setting.

**Model Training** We conduct full-parameter fine-tuning on the augmented datasets, initializing from the *Llama-3.1-8B-Instruct* checkpoint. Two separate models are trained: one for the multi-hop QA datasets and another for the KILT benchmark. The compiled multi-hop QA dataset comprises 125k training instances, whereas the KILT benchmark includes 660k instances after sub-sampling. The model is fine-tuned for 1 epoch with a maximum sequence length of 3k tokens. For the KILT benchmark, we fine-tune an E5-Mistral retriever [35] and a RankLLaMA re-ranker [24] on the respective training set to boost the ranking quality.

Further implementation details are provided in Appendix A.

Table 1: Results on multi-hop QA datasets. We report the performance of CoRAG-8B using various decoding strategies and retrieval chain lengths $L$. The "Few-shot w/o Retrieval" configuration utilizes only QA pairs without retrieval augmentation. Both DRAG and IterDRAG are based on Gemini 1.5 Flash [31], while Search-o1-32B is based on QwQ [37] and the Bing Search API.

| | 2WikiQA | | HotpotQA | | Bamboogle | | MuSiQue | |
|---|---|---|---|---|---|---|---|---|
| | EM | F1 | EM | F1 | EM | F1 | EM | F1 |
| *Few-shot w/o Retrieval* | | | | | | | | |
| 3-shot Llama-3.1-8B-Inst. | 27.6 | 32.1 | 20.8 | 28.8 | 17.6 | 21.3 | 3.4 | 9.7 |
| 3-shot GPT-4o | 39.5 | 47.3 | 38.2 | 51.2 | 49.6 | 61.5 | 15.8 | 27.2 |
| *w/ Retrieval* | | | | | | | | |
| 3-shot Llama-3.1-8B-Inst. | 30.7 | 39.9 | 34.1 | 46.6 | 28.0 | 37.3 | 7.7 | 15.4 |
| 3-shot GPT-4o | 49.0 | 56.2 | 45.8 | 59.4 | 53.6 | 63.8 | 15.7 | 25.8 |
| Self-RAG-7B | 12.2 | 24.1 | 16.6 | 29.4 | 5.6 | 16.8 | 4.6 | 13.2 |
| ITER-RETGEN | 35.5 | 47.4 | 45.1 | 60.4 | 40.0 | 50.7 | 26.1 | 42.0 |
| DRAG (32k) | 45.9 | 53.7 | 46.9 | 60.3 | 48.8 | 59.2 | 15.4 | 26.0 |
| IterDRAG (32k) | 44.3 | 54.6 | 38.3 | 49.8 | 46.4 | 56.2 | 12.5 | 23.1 |
| Search-o1-32B | 58.0 | 71.4 | 45.2 | 57.3 | **56.0** | 67.8 | 16.6 | 28.2 |
| Fine-tuned Llama-8B w/ E5$_{large}$ | 55.1 | 60.7 | 50.3 | 63.5 | 40.8 | 53.7 | 17.4 | 28.1 |
| CoRAG-8B (Ours) | | | | | | | | |
| ▷ $L$=1, greedy | 56.5 | 62.3 | 50.1 | 63.2 | 37.6 | 51.4 | 18.6 | 29.3 |
| ▷ $L$=6, greedy | 70.6 | 75.5 | 54.4 | 67.5 | 48.0 | 63.5 | 27.7 | 38.5 |
| ▷ $L$=6, best-of-4 | 71.7 | 76.5 | 55.3 | 68.5 | 51.2 | 63.1 | 28.1 | 39.7 |
| ▷ $L$=6, tree search | 71.7 | 76.4 | 55.8 | 69.0 | 48.8 | 64.4 | 29.0 | 40.3 |
| ▷ $L$=10, best-of-8 | **72.5** | **77.3** | **56.3** | **69.8** | 54.4 | **68.3** | **30.9** | **42.4** |

## 4.2 Main Results

**Multi-hop QA** In Table 1, we present a comparative analysis of CoRAG-8B against several models, including few-shot Llama-3.1-8B-Instruct [5], GPT-4o [10], Self-RAG-7B [1], ITER-RETGEN [30], DRAG, IterDRAG [42], and Search-o1-32B [22]. For a fair comparison, we also include a fine-tuned Llama-8B baseline utilizing the E5-large retriever, which is fine-tuned on the same datasets as CoRAG-8B but without retrieval chain augmentation. CoRAG-8B substantially surpasses all baselines, with the exception of the Bamboogle dataset, despite being based on a weaker LLM compared to Search-o1-32B and IterDRAG. Conversely, we recognize that fine-tuning on multi-hop QA datasets creates an advantage for CoRAG-8B, compared to the few-shot setting for DRAG and IterDRAG.

The Bamboogle dataset comprises only 125 instances, resulting in considerable variance in performance across different runs. Certain questions within Bamboogle necessitate access to knowledge more recent than the Wikipedia dump used for retrieval. Systems like Search-o1-32B, which rely on commercial search engines, possess an advantage in this regard.

**KILT Benchmark** We present several strong systems on the KILT benchmark in Table 2, including KILT-RAG [27], SEAL [2], Atlas-11B [11], RA-DIT 65B [23], and FiD with RS [9]. For submission to the KILT leaderboard, we choose the best decoding configuration for each task based on the public validation set. The results of different decoding strategies are detailed in Appendix Table 7. Our CoRAG-8B model achieves a new state-of-the-art performance across all tasks, with the exception of FEVER, where it marginally trails behind a larger model with 11B parameters.

## 4.3 Scaling Test-Time Compute

In alignment with OpenAI o1 [12], our model allows for scaling test-time compute to potentially achieve better performance without updating model weights. There are multiple ways to control the test-time compute. In Figure 3, we concentrate on two factors: the retrieval chain length $L$ and the number of sampled chains $N$ for best-of-$N$ sampling. Greedy decoding is a special instance of best-of-$N$ sampling with $N = 1$ and the temperature set to $0$.

Table 2: The downstream results on the *hidden test set* of the KILT benchmark. All scores are sourced directly from the official leaderboard, with the exception that "RA-DIT 65B" is from the original paper [23]. ∗: "Previous Best" refers to the highest score for each task on the public KILT leaderboard as of January 10, 2025.

| System | Entity Linking | | | Slot Filling | | Open QA | | | Fact |
|---|---|---|---|---|---|---|---|---|---|
| | AIDA | WnWi | WnCw | T-REx | zsRE | NQ | HoPo | TQA | FEVER |
| KILT-RAG | 72.6 | 48.1 | 47.6 | 59.2 | 44.7 | 44.4 | 27.0 | 71.3 | 86.3 |
| SEAL | - | - | - | 83.6 | 74.6 | 53.7 | 40.5 | 70.9 | 89.5 |
| Atlas-11B | 90.6 | - | - | 85.1 | 80.8 | 61.3 | 50.6 | 84.0 | **93.5** |
| RA-DIT 65B | 80.5 | - | - | 72.8 | 78.1 | 43.5 | 36.6 | 72.8 | 86.9 |
| FiD with RS | - | - | - | 85.2 | 83.7 | 61.2 | 39.1 | 84.6 | 92.2 |
| Previous Best* | 90.6 | 87.4 | 71.2 | 87.7 | 85.3 | 62.3 | 50.6 | 84.6 | **93.5** |
| CoRAG-8B (Ours) | **93.9** | **88.2** | **76.7** | **88.0** | **87.2** | **63.1** | **60.6** | **88.3** | 93.1 |

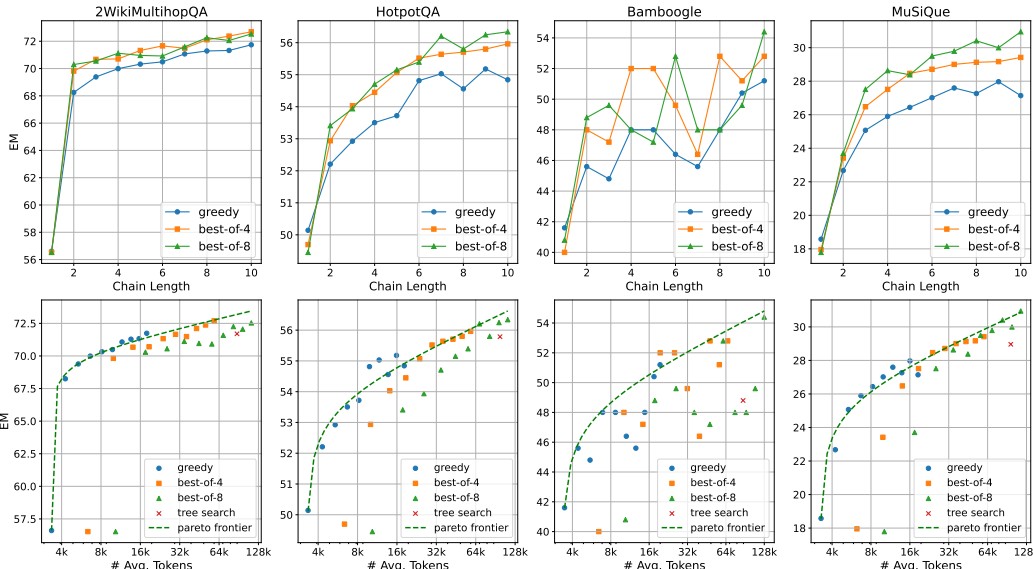

Figure 3: Scaling test-time compute on multi-hop QA datasets. The Pareto frontier is in the form of $y = a \times \log(x + b) + c$ fitted on the Pareto optimal points. A point is considered *Pareto optimal* if no other point achieves a higher EM score with less token consumption. The metric "# Avg. Tokens" represents the average number of tokens consumed per test instance, summing up both the prompt and generated tokens.

We observe that increasing the retrieval chain length $L$ results in substantial performance improvements when $L$ is small, but the gains diminish as $L$ increases. This observation aligns with the intuition that longer chains can encapsulate more reasoning steps and allows for trial-and-error exploration of various query rewriting strategies. Several examples are provided in Appendix Table 11. Conversely, increasing $N$ for best-of-$N$ sampling yields mixed effects depending on the dataset. For the most challenging dataset, MuSiQue, in terms of EM score, a larger $N$ enhances performance, whereas for the less challenging dataset, 2WikiMultihopQA, a smaller $N$ suffices. We defer the further exploration of tree search to future work, as it is considerably more computationally expensive than greedy decoding and best-of-$N$ sampling.

The Pareto frontier between the EM score and token consumption approximately follows a log-linear trajectory for up to 128k tokens, although the scaling behavior varies across different datasets. This observation assists practitioners in making informed decisions regarding the allocation of test-time compute based on the quality requirements. It is important to note that we make several simplifications

in this scaling study, such as treating the prompt tokens equivalently to the generated tokens and ignoring the retrieval costs. A more rigorous analysis could take these factors into account.

## 5   Analysis

Table 3: Ablation study results. "Iterative training" employs a trained CoRAG model for another round of rejection sampling. "Distill from GPT-4o" leverages the GPT-4o model to generate retrieval chains. "Weak-to-strong Generalization" utilizes weaker LLMs for retrieval chain generation while using stronger LLMs (*Llama-3.1-8B-Inst.*) for training. "Different Retrievers" replaces the text retriever at test time.

|  | 2WikiQA | | HotpotQA | | Bamboogle | | MuSiQue | |
|---|---|---|---|---|---|---|---|---|
|  | EM | F1 | EM | F1 | EM | F1 | EM | F1 |
| CoRAG-8B (L=6, greedy) | 70.6 | 75.5 | 54.4 | 67.5 | 48.0 | 63.5 | 27.7 | **38.5** |
| ▷ iterative training | 72.2 | 76.9 | 53.4 | 66.5 | 45.6 | 60.9 | 26.6 | 37.6 |
| ▷ distill from GPT-4o | **75.1** | **79.5** | **56.6** | **70.2** | **51.2** | **67.0** | **28.2** | **38.5** |
| *Weak-to-strong Generalization* | | | | | | | | |
| w/ Llama-3.2-1B-Inst. | 59.3 | 64.2 | 50.3 | 63.6 | 40.8 | 51.6 | 22.3 | 32.7 |
| w/ Llama-3.2-3B-Inst. | 69.9 | 74.0 | 53.9 | 67.3 | 45.6 | 59.8 | 25.2 | 36.0 |
| *Different Retrievers* | | | | | | | | |
| E5-base w/o chain-of-retrieval | 53.1 | 58.9 | 47.9 | 61.1 | 38.4 | 52.7 | 15.8 | 26.4 |
| ▷ L=6, best-of-4 | 70.8 | 75.4 | 53.0 | 66.2 | 47.2 | 59.8 | 26.3 | 37.6 |
| BM25 w/o chain-of-retrieval | 49.1 | 55.3 | 46.9 | 60.3 | 36.8 | 48.6 | 14.3 | 24.8 |
| ▷ L=6, best-of-4 | 62.6 | 67.7 | 51.6 | 64.7 | 37.6 | 52.5 | 23.5 | 33.0 |

### 5.1   Iterative Rejection Sampling

Our framework facilitates self-improvement through iterative training, akin to the iterative rejection sampling employed in LLM post-training [5]. By utilizing the same prompt templates for both data generation and model training, a trained CoRAG model can generate new sets of retrieval chains. However, the results in Table 3 are mixed, showing performance improvements on the 2WikiMultihopQA dataset but slight declines on other datasets. This indicates that instruction-tuned LLMs already possess a strong ability to generate high-quality retrieval chains.

### 5.2   Robustness and Generalization

**Different Retrievers** We further investigate the influence of various text retrievers at test time. Instead of using the E5-large dense retriever, we substitute it with two weaker alternatives in a plug-and-play fashion: E5-base and BM25. Across all datasets, we observe consistent performance gains when investing more test-time compute, although stronger retrievers continue to outperform in terms of absolute performance. Improvements to text retriever quality represent an orthogonal dimension that can further amplify CoRAG's performance gains.

**Weak-to-strong Generalization** Due to the need of repeated sampling and autoregressive generation, the retrieval chain generation process costs more GPU hours than the model training. To mitigate this cost, one strategy is to employ weaker LLMs for retrieval chain generation and subsequently fine-tune stronger LLMs on the augmented datasets, similar to the weak-to-strong generalization setting [4].

The results in Table 3 demonstrate that utilizing Llama-3B achieves very close performance compared to the 8B model, whereas Llama-1B exhibits a noticeable performance drop. Manual inspection reveals that the 1B model frequently struggles to follow the given instructions, resulting in sub-optimal retrieval chains. Employing weaker LLMs also lowers the barrier to adopting more computationally expensive tree search strategies during data generation, which show great potential in mathematical reasoning tasks [7]. In contrast, distilling from a stronger model like GPT-4o yields a further performance boost, indicating that the quality of the retrieval chains is crucial for the final performance.

## 5.3 Does Chain-of-Retrieval Always Help?

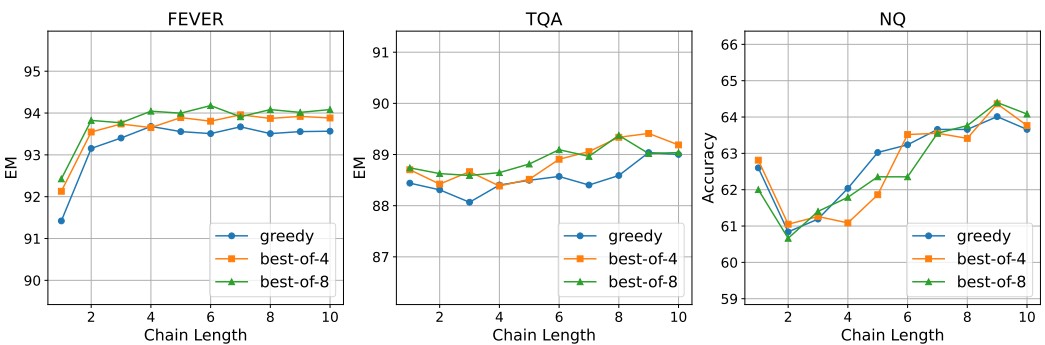

Figure 4: Scaling test-time compute across three datasets from the KILT benchmark. We report scores on the public validation set.

Multi-hop QA datasets are specifically designed to evaluate complex reasoning capabilities and are expected to benefit from the chain-of-retrieval mechanism. Table 1 presents empirical evidence supporting this assertion. In contrast, for tasks that a single retrieval step is typically sufficient, the advantage tends to be marginal, as demonstrated in Figure 4. Datasets such as NQ [19] and TriviaQA [17] are known for their (mostly) single-hop nature. This phenomenon implies that decoding strategies should be adaptive based on the complexity of the query. Additional results on the full KILT benchmark are listed in Appendix Table 7, where similar observations for other task types also hold.

## 5.4 Learning to Stop at Test Time

Instead of always performing $L$ retrieval steps, we explore a model variant that learns to stop at test time. After each retrieval step, the model is prompted to predict whether the information gathered thus far suffices to answer the query. Note that this prompt itself also incurs token consumption and additional cost. The decoding space is constrained to two tokens: *"Yes"* and *"No"*. If the decoded output is "*Yes*", no further sub-queries are generated. By adjusting the logit bias of the "*Yes*" token, we can control the early stopping behavior.

During the training phase, an additional loss term is added for the stop prediction task. The target output is "*Yes*" if the current retrieval chain encompasses the prefix that maximizes the likelihood of the final answer, and "*No*" otherwise. The associated prompt template is in Appendix Section D.

In Figure 5, we illustrate how the performance varies along with the token consumption on the MuSiQue dataset. While early stopping can save some amount of token quota, it comes at the cost of performance degradation. The optimal configuration depends on the dataset characteristics and the quality expectations.

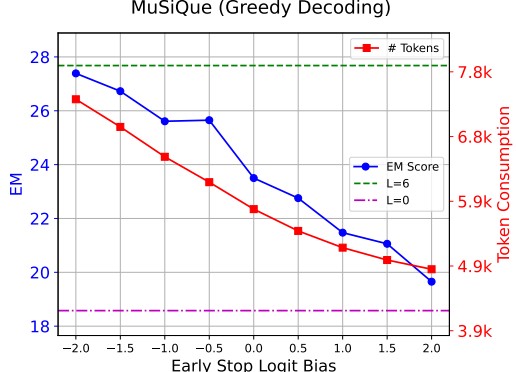

Figure 5: Learning to stop at test time. Larger logit bias values result in earlier stopping. $L = 6$ correspond to always performing 6 retrieval steps, while $L = 0$ indicate no intermediate retrieval steps.

## 5.5 Does CoRAG Learn to Retrieve Better?

To evaluate whether CoRAG improves retrieval quality beyond just answer accuracy, we measure retrieval recall across multiple datasets. We report Recall@k metrics for $k \in \{10, 20, 100\}$, comparing standard retrieval using $E5_{large}$ against our approach. We follow the evaluation protocol from DPR [18] for calculating recall based on answer matches, as not all datasets provide gold supporting paragraphs. For CoRAG, we utilize reciprocal rank fusion to merge multiple retrieval results from the chain into a single ranked list, from which recall is calculated.

Table 4: Retrieval recall comparison between standard retrieval and CoRAG across multi-hop QA datasets.

|  | R@10 | R@20 | R@100 |
|---|---|---|---|
| HotpotQA w/ $E5_{large}$ | 59.1 | 65.2 | 76.8 |
| w/ CoRAG | **72.1** | **76.7** | **84.3** |
| 2WikiMultiHopQA w/ $E5_{large}$ | 54.9 | 62.1 | 74.6 |
| w/ CoRAG | **81.4** | **84.8** | **88.8** |
| Bamboogle w/ $E5_{large}$ | 31.2 | 40.0 | 57.6 |
| w/ CoRAG | **59.2** | **68.0** | **75.2** |
| MuSiQue w/ $E5_{large}$ | 29.0 | 36.5 | 52.7 |
| w/ CoRAG | **47.1** | **54.6** | **68.4** |

The results in Table 4 demonstrate that CoRAG consistently improves recall across all datasets and recall thresholds. The improvements are particularly pronounced on more challenging datasets like MuSiQue and Bamboogle, where single-step retrieval struggles most. This indicates that CoRAG's iterative query reformulation and decomposition strategy effectively addresses the limitations of traditional dense retrieval, enabling the model to gather more relevant information through multiple retrieval steps.

## 6 Conclusion

In this work, we introduce CoRAG, a framework that teaches LLMs to conduct iterative retrieval and reasoning to answer complex queries. The intermediate retrieval chains are automatically generated via rejection sampling, thereby alleviating the need for manual annotation. At test time, we offer multiple decoding strategies to manage the trade-off between performance and compute. Our experiments demonstrate that CoRAG-8B achieves state-of-the-art performance on both multi-hop QA datasets and the KILT benchmark, surpassing many baselines built with larger LLMs. A comprehensive analysis is conducted to understand its scaling behavior and generalization capability. In the future, we intend to extend CoRAG to more challenging and economically valuable RAG tasks, advancing towards building factual and trustworthy AI systems.

## 7 Limitations and Broader Impacts

This study primarily investigates RAG tasks characterized by short and easy-to-verify answers, such as multi-hop QA and entity linking. However, real-world applications often necessitate addressing more complex tasks that demand generating long-form outputs. A significant challenge in long-form generation lies in the absence of robust evaluation metrics within the current research landscape.

Regarding broader impacts, the proposed framework aims to improve the factuality and groundedness of language model outputs. It is anticipated that this work can facilitate more efficient and effective information retrieval for users. Nevertheless, the inherent risk of hallucination persists and warrants careful monitoring in practical deployments.

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

## A  Implementation Details

**Rejection Sampling** For each training instance, we sample up to 16 retrieval chains, with the maximum length randomly selected from the interval $[1, 5]$. The sampling temperature is set to 0.7 for sub-query generation and 0 for sub-answer generation. Chain generation is terminated if the sub-answer matches the correct answer or if the average conditional log-likelihood of the correct

answer exceeds $-0.05$. For each sub-query, we utilize the E5-large retriever [2] to retrieve the top-5 most relevant documents from the KILT version of the Wikipedia corpus [25]. This corpus comprises 36 million passages.

Table 5: Hyperparameters for training CoRAG.

|  | Multi-hop QA | KILT Benchmark |
|---|---|---|
| Initialization | *Llama-3.1-8B-Instruct* | |
| Learning rate | $5 \times 10^{-6}$ | $10^{-5}$ |
| Batch size | 256 | 1024 |
| Epoch | 1 | 1 |
| Warmup steps | 100 | 100 |
| # Training samples | $125k$ | $660k$ |
| # Retrieved passages | 20 | 20 |
| Max sequence length | 3072 | 3072 |

Table 6: Statistics of the datasets used for multi-hop QA training.

|  | 2WikiMultihopQA | HotpotQA | Bamboogle | MuSiQue |
|---|---|---|---|---|
| # Training Samples | $15,000$ | $90,447$ | - | $19,938$ |
| # Validation Samples | $12,576$ | $7,405$ | 125 | $2,417$ |

**Multi-Hop QA Training Hyperparameters** The training set is the union of the 2WikiMultihopQA, HotpotQA, and MuSiQue datasets, comprising a total of 125k samples as shown in Table 6. The Bamboogle dataset, consisting of only 125 questions, is reserved for evaluation only. Additional hyperparameters are detailed in Table 5. To balance the three loss terms in Section 3.2, we set a sample ratio of 0.2 for both the sub-query and sub-answer generation tasks; this ratio is also applied to the KILT training.

**KILT Training Hyperparameters** We utilize the official training set of the KILT benchmark, omitting the ELI5 and WoW datasets due to the lack of reliable evaluation metrics. To balance the task distribution, we only select 100k samples for large datasets like T-REx and Zero-Shot RE. In accordance with the benchmark's guidelines, we also add 100k samples from the BLINK dataset for entity linking.

Rather than using off-the-shelf retrievers, we fine-tune an E5-Mistral retriever following Wang et al., and a RankLLaMA re-ranker following Ma et al.. We adhere to the exact training hyperparameters outlined in the original papers, except that the training data is replaced with the KILT training set. For training the RankLLaMA re-ranker, the backbone is initialized with the *Llama-3-8B-Base* model, as opposed to Llama-2, to enhance performance. Retrieval and re-ranking scores are presented in Table 8.

All training jobs are conducted using 8 A100 GPUs. The multi-hop QA task requires less than 6 hours of training, whereas the KILT training takes approximately 30 hours. When submitting to the KILT leaderboard, we select the optimal decoding strategy for each task based on validation set performance.

**Decoding Strategies** In the context of best-of-$N$ sampling, the temperature is set to 0.7 for sub-query generation. For sub-answer generation and final answer prediction, the temperature is always set to 0 across all decoding strategies. Regarding tree search, we set the expansion size to 4 and the number of rollouts to 2. Given that tree search incurs a significantly higher token consumption compared to other decoding strategies, we limit the rollouts to a maximum of 2 steps for each expansion. To avoid the model from generating repetitive sub-queries endlessly, any generated sub-query identical to previous ones is discarded.

**Evaluation** For multi-hop QA tasks, we evaluate the performance using the exact match (EM) and F1 scores [18]. For Self-RAG-7B, we reproduce the results utilizing the FlashRAG [16] toolkit with the official checkpoint released by the authors.

---

[2] https://huggingface.co/intfloat/e5-large-v2

For the KILT benchmark, we employ the official evaluation scripts provided by the organizers. For Open QA tasks, the main evaluation metric is the EM score, while other task types are evaluated using accuracy scores. The KILT benchmark also offers a variant of the evaluation protocol that requires the model not only to generate the correct answer but also to provide the correct supporting evidence. However, our method spreads the evidence documents across the retrieval chain, rendering it challenging to conform to such an evaluation protocol.

# B  Additional Results

Table 7: Downstream results on the public *validation set* of the KILT benchmark.

| System | Entity Linking | | | Slot Filling | | Open QA | | | Fact |
|---|---|---|---|---|---|---|---|---|---|
| | AIDA | WnWi | WnCw | T-REx | zsRE | NQ | HoPo | TQA | FEVER |
| CoRAG-8B (Ours) | | | | | | | | | |
| ▷ $L$=1, greedy | 90.4 | 86.0 | **76.8** | **87.0** | 82.1 | 62.5 | 56.4 | 88.4 | 91.4 |
| ▷ $L$=6, greedy | **92.7** | **87.4** | 75.8 | 86.6 | **83.8** | **63.2** | 59.1 | 88.6 | 93.8 |
| ▷ $L$=6, best-of-4 | 92.5 | **87.4** | 75.8 | 86.3 | 83.5 | 62.6 | 59.6 | 88.7 | **93.9** |
| ▷ $L$=6, tree search | 91.8 | 86.8 | 75.5 | 86.4 | 83.0 | 62.4 | **59.9** | **88.9** | **93.9** |

Table 8: Retrieval results (R-Precision) on the public *validation set* of the KILT benchmark. For re-ranking, we use the top-100 candidates from the fine-tuned retriever as input.

| System | Entity Linking | | | Slot Filling | | Open QA | | | Fact |
|---|---|---|---|---|---|---|---|---|---|
| | AIDA | WnWi | WnCw | T-REx | zsRE | NQ | HoPo | TQA | FEVER |
| Fine-tuned E5$_{\text{mistral}}$ | 92.9 | 86.7 | 76.0 | 80.5 | 95.3 | 77.7 | 66.7 | 78.9 | 90.9 |
| ▷ w/ re-ranking | 93.3 | 88.0 | 77.1 | 83.2 | 97.6 | 78.2 | 78.2 | 81.5 | 92.3 |

**Different Decoding Strategies on the KILT Benchmark** In Table 7, we present the results of various decoding strategies applied to the *validation set* of the KILT benchmark. Given that most tasks within the KILT benchmark are much easier for strong dense retrievers compared to multi-hop QA, the disparity in performance across different decoding strategies is less pronounced. This observation underscores the necessity of developing a system capable of adaptively selecting the optimal decoding strategy to effectively balance the trade-off between performance and test-time compute.

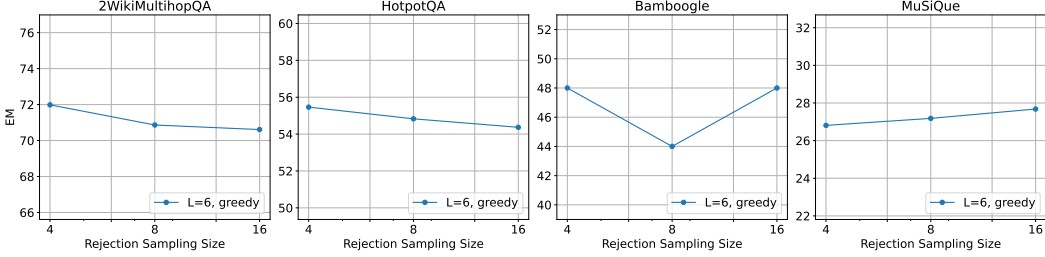

Figure 6: Scaling rejection sampling compute for training data generation. We vary the number of sampled chains from 4 to 16 while maintaining all other hyperparameters fixed.

**Scaling Compute for Training Data Generation** Within our proposed framework, rather than investing more compute at test time, we can scale the compute for retrieval chain generation during rejection sampling. By increasing the number of sampled chains, we may identify better chains that contribute to higher-quality training data. However, as illustrated in Figure 6, no definitive trend emerges indicating that increasing the number of sampled chains always leads to better performance. Conversely, the training loss consistently decreases as we scale up rejection sampling, suggesting that the training data becomes less noisy and easier to fit. We hypothesize that the majority of sampled

chains are already of high quality and that LM fine-tuning exhibits considerable robustness to noisy training data.

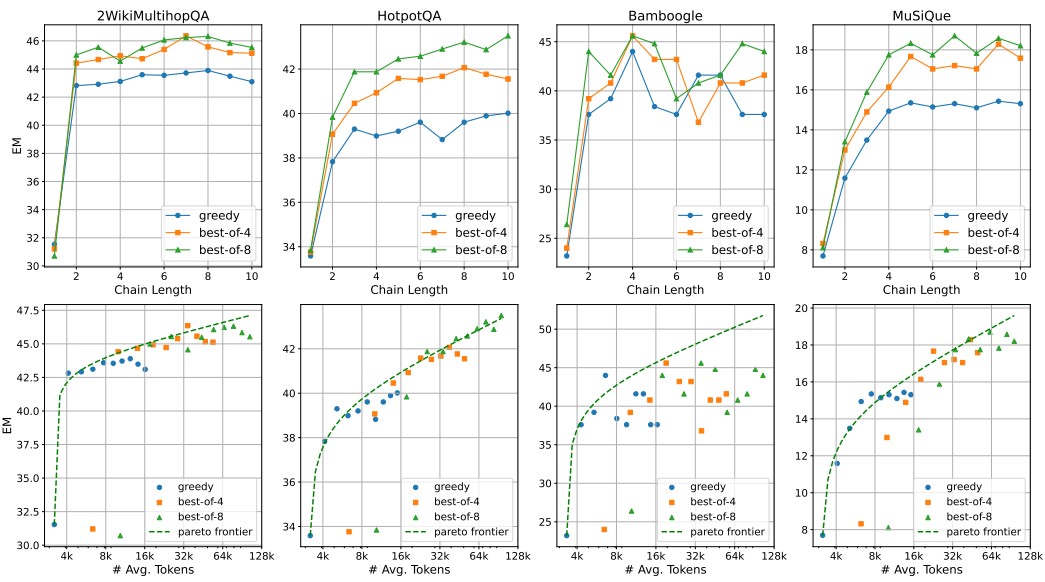

Figure 7: Scaling test-time compute on multi-hop QA datasets with *Llama-3.1-8B-Instruct*. No fine-tuning is performed on the model weights.

**Scaling Test-Time Compute without Model Fine-Tuning** In Figure 7, we present the scaling results on multi-hop QA datasets using the *Llama-3.1-8B-Instruct* model directly without any fine-tuning. The scaling curves are similar to those observed in Figure 3, but the absolute performance is significantly lower, indicating that targeted fine-tuning is essential for improving the scaling upper bound.

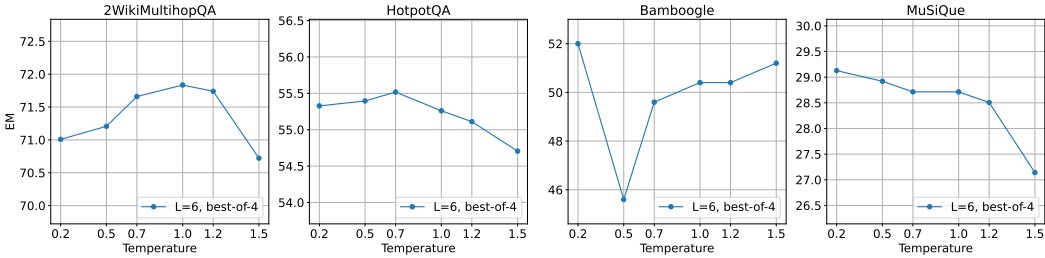

Figure 8: Effects of varying the sampling temperature on multi-hop QA datasets.

**Effects of Sampling Temperature** In best-of-$N$ sampling, the sampling temperature controls the diversity and quality trade-off in the generated retrieval chains. A higher temperature results in more diverse chains, albeit with the potential introduction of increased noise. Figure 8 illustrates the lack of a consistent conclusion regarding the impact of sampling temperature on performance. For the MuSiQue and HotpotQA datasets, a lower temperature generally yields superior results, whereas for the 2WikiMultihopQA dataset, a medium temperature leads to the best performance. As a result, we stick to a temperature of $0.7$ for both rejection sampling and test-time decoding for simplicity.

**Extension to Other Model Families** To demonstrate that our CoRAG framework is agnostic to model families and not limited to Llama-based architectures, we conduct experiments using Qwen3-4B and Qwen3-8B [38] models following the same training procedure. As shown in Table 9, CoRAG consistently outperforms the baseline fine-tuned models across all datasets and model sizes, with improvements of over 10 EM points on average. This validates that the chain-of-retrieval mechanism

Table 9: Extension to the Qwen3 model families.

| | 2WikiQA | | HotpotQA | | Bamboogle | | MuSiQue | |
|---|---|---|---|---|---|---|---|---|
| | EM | F1 | EM | F1 | EM | F1 | EM | F1 |
| Fine-tuned Qwen3-4B w/ E5$_{large}$ | 49.3 | 55.3 | 45.0 | 57.9 | 32.8 | 43.1 | 13.4 | 23.8 |
| CoRAG-Qwen3-4B (L=6, greedy) | **69.3** | **74.1** | **51.6** | **64.2** | **49.6** | **62.5** | **24.0** | **34.5** |
| Fine-tuned Qwen3-8B w/ E5$_{large}$ | 52.1 | 57.9 | 47.1 | 60.0 | 33.6 | 47.6 | 15.3 | 26.3 |
| CoRAG-Qwen3-8B (L=6, greedy) | **70.0** | **74.8** | **52.8** | **66.0** | **49.6** | **63.7** | **25.2** | **35.9** |

is broadly applicable across different model architectures and confirms the generalizability of our approach beyond specific model families.

**Case Analysis** Table 11 presents several model predictions on the validation set of the HotpotQA dataset. We compare the performance of RAG without chain-of-retrieval against CoRAG. CoRAG effectively decompose the complex multi-hop queries into a sequences of simpler sub-queries and dynamically conducts query reformulation when the retrieved information proves unhelpful. In the fourth example, the model initially hallucinates some incorrect information but subsequently self-corrects by verifying the poet's name and country of origin through additional retrieval steps.

## C   Statistical Significance

We compute the 95% confidence intervals for our main results in Table 1 and 2 using the bootstrap resampling method. On all datasets except the Bamboogle dataset, we observe that the performance of CoRAG-8B is significantly better than the baselines.

## D   Prompts

Table 10: Task descriptions for each dataset.

| Dataset | Task Description |
|---|---|
| HotpotQA / 2WikiMulti-hopQA | answer multi-hop questions |
| NQ | answer natural questions from Google search |
| AidaYago 2 / WnWi / WnCw / Blink | link the mention surrounded by [START_ENT] and [END_ENT] to the title of the correct Wikipedia page |
| FEVER | verify if the claim is supported or refuted |
| T-REx / Zero-Shot RE | given head entity and relation separated by [SEP], find the correct tail entity, return the title of its Wikipedia page |
| Trivia QA | answer trivia questions |
| MuSiQue / Bamboogle | answer multi-hop questions |

All prompts are listed in this section. Texts highlighted in blue denote placeholders to be filled in. The "*task description*" field is dataset-specific, with corresponding values detailed in Table 10.

Table 11: Examples from the validation set of the HotpotQA dataset. For conciseness, all retrieved documents at each step are omitted. Correct answers are highlighted in blue, while incorrect answers are highlighted in red.

---

**Query:** What wrestling team is Mathew Thomas Rehwoldt a part of?

*RAG without Chain-of-Retrieval*
Final Answer: **WWE ✗**

*CoRAG (Ours)*
Sub-query 1: What is Mathew Thomas Rehwoldt's profession?
Sub-answer 1: No relevant information found.
Sub-query 2: What is Mathew Thomas Rehwoldt's name in the wrestling industry?
Sub-answer 2: Aiden English
Sub-query 3: What wrestling team is Aiden English a part of?
Sub-answer 3: The Vaudevillains
Final Answer: **The Vaudevillains ✓**

---

**Query:** How many months apart are Johan Mjällby and Neil Lennon in age?

*RAG without Chain-of-Retrieval*
Final Answer: **two months ✗**

*CoRAG (Ours)*
Sub-query 1: What is Johan Mjällby's birthdate?
Sub-answer 1: 9 February 1971
Sub-query 2: What is Neil Lennon's birthdate?
Sub-answer 2: 25 June 1971
Sub-query 3: What is the difference in months between 9 February 1971 and 25 June 1971?
Sub-answer 3: 4 months
Final Answer: **4 ✓**

---

**Query:** Thomas Parker Sanborn was a model for the protagonist in the only novel by a poet originally from what country?

*RAG without Chain-of-Retrieval*
Final Answer: **Italian ✗**

*CoRAG (Ours)*
Sub-query 1: What country is the poet originally from?
Sub-answer 1: Hungary, Romania, Switzerland, China, Ukraine, France, Turkey.
Sub-query 2: What is the name of the poet who wrote the only novel?
Sub-answer 2: Emily Brontë
Sub-query 3: What country is the poet who wrote the only novel originally from?
Sub-answer 3: United States
Sub-query 4: What is the name of the poet who wrote the only novel that Thomas Parker Sanborn was a model for?
Sub-answer 4: George Santayana
Sub-query 5: What poet wrote the only novel that Thomas Parker Sanborn was a model for?
Sub-answer 5: George Santayana
Sub-query 6: What country is the poet George Santayana originally from?
Sub-answer 6: Spain.
Final Answer: **Spain ✓**

---

**Query:** Which university is further North, University of New Hampshire or Stony Brook University?

*RAG without Chain-of-Retrieval*
Final Answer: **University of New Hampshire ✗**

*CoRAG (Ours)*
Sub-query 1: What is the location of the University of New Hampshire?
Sub-answer 1: Durham, New Hampshire.
Sub-query 2: What is the location of Stony Brook University?
Sub-answer 2: Stony Brook, NY.
Sub-query 3: What is the latitude of Durham, New Hampshire?
Sub-answer 3: No relevant information found.
Sub-query 4: What is the latitude of Stony Brook, NY?
Sub-answer 4: No relevant information found.
Final Answer: **University of New Hampshire ✗**

---

### Prompt: Sub-query Generation

You are using a search engine to answer the main query by iteratively searching the web. Given the following intermediate queries and answers, generate a new simple follow-up question that can help answer the main query. You may rephrase or decompose the main query when previous answers are not helpful. Ask simple follow-up questions only as the search engine may not understand complex questions.

## Previous intermediate queries and answers
{intermediate queries and answers}

## Task description
{task description}

## Main query to answer
{query}

Respond with a simple follow-up question that will help answer the main query, do not explain yourself or output anything else.

---

### Prompt: Intermediate Answer Generation

Given the following documents, generate an appropriate answer for the query. DO NOT hallucinate any information, only use the provided documents to generate the answer. Respond "No relevant information found" if the documents do not contain useful information.

## Documents
{retrieved documents}

## Query
{sub-query}

Respond with a concise answer only, do not explain yourself or output anything else.

---

### Prompt: Final Answer Generation

Given the following intermediate queries and answers, generate a final answer for the main query by combining relevant information. Note that intermediate answers are generated by an LLM and may not always be accurate.

## Documents
{retrieved documents}

## Intermediate queries and answers
{intermediate queries and answers}

## Task description
{task description}

## Main query
{query}

Respond with an appropriate answer only, do not explain yourself or output anything else.

## Prompt: Learning to Stop

Given the following intermediate queries and answers, judge whether you have enough information to answer the main query. If you believe you have enough information, respond with "Yes", otherwise respond with "No".

## Intermediate queries and answers
{intermediate queries and answers}

## Main query
{query}

Respond with "Yes" or "No" only, do not explain yourself or output anything else.

