# OpenReview forum: "Chain-of-Retrieval Augmented Generation"
_NeurIPS.cc/2025/Conference — NeurIPS 2025 poster_

### Official Review · Reviewer_PDbv · 2025-06-25

**Clarity:** 3
**Significance:** 3
**Originality:** 2
**Rating:** 4
**Confidence:** 3

**Summary:**

This paper introduces an approach for training o1-like RAG models called CoRAG. It combines the RAG with Chain-of-Thought during training. CoRAG first performs rejection sampling with log-likelihood as the signal of the correction, then training for the next sub-query prediction, sub-answer prediction, and final answer prediction. This method also introduces inference strategies: Greedy Decoding, Best-of-N Sampling, and Tree Search. Experiments show good performance.

**Questions:**

I am a little confused about the terms "Current State" and "Next Action" in Figure 1. How many times will the LLM be trained? Does the "State" refer to the token?

Some recent papers have proposed R1-Zero-like RAG methods. It would be interesting to compare them. I am aware that they should be considered contemporary work, and this will not influence my evaluation of this paper.

**Ethical Concerns:**

["NO or VERY MINOR ethics concerns only"]

**Final Justification:**

Good writing and experiments. The method is straightforward, and I am a little concerned about the novelty. So my rating is 4.

**Limitations:**

yes

**Quality:**

3

**Strengths And Weaknesses:**

The paper is well-written and easy to follow.

The main contribution is to put RAG in o1-like training. It is interesting to combine the o1-training manner and RAG. However, the only novelty seems to be "add RAG" (please let me know if the authors believe this is incorrect). The method is also straightforward, but a simple method could also be a good method.

Experiments are conducted on two sets of benchmarks with several datasets, which demonstrates its strong performance as well as good robustness and generalization.

In summary, the method is straightforward but useful, as demonstrated by experiments. I am slightly positive about the paper.

---

> ### Author Rebuttal · Authors · 2025-07-31
>
> Thank you for your positive feedback and constructive suggestions!
>
> >"I am a little confused about the terms "Current State" and "Next Action" in Figure 1. How many times will the LLM be trained? Does the "State" refer to the token?"
>
> In Figure 1, "Current State" denotes the input context and instructions provided to the LLM. The "Next Action" refers to the LLM output responding to the given instruction.
> To illustrate, consider sub-query generation: the "Current State" encompasses the original query along with all historical sub-queries and their corresponding answers, while the "Next Action" is the newly generated sub-query. This conceptualization similarly applies to both sub-answer generation and final answer generation. For detailed examples of our prompt templates, please refer to Page 19 of the manuscript.
>
> To clarify the training process, the LLM is trained only once. We mix the training data for all three tasks, enabling simultaneous multi-task learning.
>
> >"Some recent papers have proposed R1-Zero-like RAG methods. It would be interesting to compare them. I am aware that they should be considered contemporary work, and this will not influence my evaluation of this paper."
>
> We appreciate your suggestion! We have incorporated a brief discussion of Search-R1 in the related work section. RL holds great promises for agentic systems; however, it is also well-known for high computational demands and training instability. Given the emergence of numerous concurrent works applying RL to RAG in recent months, we intend to substantially expand upon this discussion in the final version of our paper.

---

> > ### Comment · Reviewer_PDbv · 2025-08-03
> >
> > Thanks for the response to clarify my confusion. I will keep the score. Thanks.

---

### Official Review · Reviewer_wBfm · 2025-07-02

**Clarity:** 3
**Significance:** 3
**Originality:** 3
**Rating:** 5
**Confidence:** 4

**Summary:**

The paper introduces Chain-of-Retrieval Augmented Generation (CoRAG), a method that decomposes or reformulates a query into subqueries that the model can more effectively address. Training is conducted iteratively with rejection sampling, and the authors show how test-time scaling further enhances accuracy. CoRAG achieves strong performance, outperforming several baselines and setting a new state of the art on the KILT benchmark. The paper also offers thoughtful analysis and ablations, exploring key aspects such as the role of rejection sampling, the impact of using different retrievers or subquery generation models, and whether the model can learn to decide when to terminate the retrieval chain.

**Questions:**

It would be helpful to better understand the failure modes of the proposed approach. For instance, was there any analysis on the relevance or utility of the generated subqueries—e.g., what percentage of the chain consists of subquestions that are actually helpful or on-topic? Given that performance improves with increased generation length, one might assume that longer chains could also introduce noise or irrelevant steps. Did the authors observe any failure cases where longer chains hurt performance, or where irrelevant subquestions derailed the retrieval process?

**Ethical Concerns:**

["NO or VERY MINOR ethics concerns only"]

**Limitations:**

Yes

**Quality:**

3

**Strengths And Weaknesses:**

*Strengths*

- The paper is well-written and easy to follow.

- The proposed method achieves strong results across several tasks, and the accompanying analysis offers valuable insights that are likely to benefit the community.

- The test-time scaling experiments are particularly interesting and add depth to the empirical evaluation.

*Weaknesses*

- A central claim of the paper appears to be that training a model to decompose queries via rejection sampling leads to performance gains. However, the results suggest that rejection sampling had limited impact, which weakens this contribution. It would have been valuable to explore this direction further—e.g., through preference learning or alternative training approaches.

- Some relevant (albeit slightly older work was missed)  eg [1]


[1] Buck, Christian, et al. "Ask the right questions: Active question reformulation with reinforcement learning." arXiv preprint arXiv:1705.07830 (2017).


Overall, I think this is a decent contribution and I recommend acceptance

---

> ### Author Rebuttal · Authors · 2025-07-31
>
> Thank you for your positive feedback and constructive suggestions!
>
> > "the results suggest that rejection sampling had limited impact, which weakens this contribution. It would have been valuable to explore this direction further—e.g., through preference learning or alternative training approaches."
>
> **Our findings indicate that the choice of LLM for rejection sampling significantly impacts performance (Table 3), whereas merely increasing the number of sampled trajectories yielded limited gains (Figure 6 in Appendix B).** We hypothesize that the diversity and quality of sampled trajectories are currently constrained by the inherent capabilities of the LLM. We agree that exploring approaches like online rejection sampling or reinforcement learning, as you suggested, holds substantial promise for addressing this limitation, and we plan to investigate these directions in our future work.
>
> > "Did the authors observe any failure cases where longer chains hurt performance, or where irrelevant subquestions derailed the retrieval process?"
>
> Yes, we indeed observed common failure patterns. A prominent one is the generation of sub-queries that are mere paraphrases of previous ones. The model sometimes persists in asking similar questions even when the retriever fails to find relevant information, rather than exploring alternative avenues. This issue is particularly pronounced with smaller models such as Llama-3.2-1B. We will provide a more detailed analysis of these failure cases in the final version of the paper.
>
> > "Some relevant (albeit slightly older work was missed) eg [1]
> [1] Buck, Christian, et al. "Ask the right questions: Active question reformulation with reinforcement learning." arXiv preprint arXiv:1705.07830 (2017)."
>
> Thanks for the pointer!
> We agree that reinforcement learning is a crucial area for developing more effective RAG agents.
> We will discuss this work in the related work section.

---

### Official Review · Reviewer_gW6e · 2025-07-03

**Clarity:** 4
**Significance:** 2
**Originality:** 2
**Rating:** 3
**Confidence:** 4

**Summary:**

This paper proposes a novel framework named CoRAG for training and inference in RAG. The key idea is to perform multi-step retrieval via intermediate query reformulation and reasoning. The authors use rejection sampling to automatically generate training data with retrieval chains from QA-only datasets and train LLMs to predict sub-queries, sub-answers, and final answers. At test time, they explore several decoding strategies (e.g., greedy, best-of-N, tree search) to balance performance and efficiency. It achieves good results across multi-hop QA datasets and the KILT benchmark.

**Questions:**

Questions:

Q1. Following W2. What is the retrieval accuracy (e.g., support fact recall) of CoRAG compared to strong retrievers?

Q2. Following W3, Does the proposed framework improve performance when applied to other LLMs beyond LLaMA-3?

Q3. Is there any human evaluation to verify the validity of generated sub-questions and sub-answers?

Q4. In the ablation study on retrievers (Line 234), performance gains are attributed to stronger retrievers. Can the authors isolate the contribution of the CoRAG mechanism from that of the retrieval encoder?

**Ethical Concerns:**

["NO or VERY MINOR ethics concerns only"]

**Final Justification:**

Thank you for your response, which partially addresses my concerns regarding retrieval, so I have raised scores accordingly. However, my key concern about the unfair comparison noted in the weaknesses remains: it is still not fair to compare your fine-tuned model to other zero/few-shot LLMs, or to other fine-tuned models without retrieval.

**Limitations:**

Yes

**Quality:**

2

**Strengths And Weaknesses:**

Strengths:

1. The methodology is clearly described, and the system components are coherently integrated.

2. The test-time compute scaling analysis is well done and offers practical value for real-world deployment.


Weaknesses:
1. In Table 1, most baselines compared are designed for zero-shot or few-shot prompting, whereas the proposed CoRAG is fine-tuned for multi-hop QA task.  The only fine-tuned baseline is a standard LLaMA-8B model, which is not specifically optimized for multi-hop QA task. This discrepancy makes it challenging to draw clear conclusions from the unfair comparison. Including stronger fine-tuned baselines, especially encoder-based models optimized for multi-hop QA, would help contextualize CoRAG’s performance. For instance, CoRAG achieves an F1 of 69.8 on HotpotQA, whereas leaderboard models exceed 80.5 [1]. Similarly, on MuSiQue, CoRAG’s best F1 is 42.4, while top models score above 78.4 [2]. The full wiki setting is close to open-domain setting.
[1] https://hotpotqa.github.io/
[2] https://leaderboard.allenai.org/musique_ans/submissions/public

2. Since one of the paper’s key ideas is improving retrieval quality through dynamic retrieval, it would be helpful to include standard retrieval metrics (e.g., recall@k or supporting fact accuracy) to better assess the effectiveness of the proposed method in retrieving supporting paragraphs, which is missing from the paper.


3. The current experiments are conducted using one backbone. Including results with additional model backbones could strengthen the case for generalizability and robustness of the proposed approach.

4. The idea of multi-step, dynamic retrieval has been extensively studied in prior work, such as IRCoT, Self-RAG, ITER-RETGEN, and others.

---

> ### Author Rebuttal · Authors · 2025-07-31
>
> Thanks for your constructive feedback!
>
> >"CoRAG achieves an F1 of 69.8 on HotpotQA, whereas leaderboard models exceed 80.5 [1]. Similarly, on MuSiQue, CoRAG’s best F1 is 42.4, while top models score above 78.4 [2]"
>
> **The leaderboard scores you cited for HotpotQA and MuSiQue are based on distinct and easier settings, making them not directly comparable to our results.** Specifically:
> * The MuSiQue-Ans leaderboard and HotpotQA Distractor setting provide gold paragraphs within the context, simplifying the task to reading comprehension. In contrast, our work focuses on an open-domain QA setting, where no gold paragraphs are assumed.
> * The HotpotQA Full-Wiki setting, by design, only requires retrieval from the abstracts of Wikipedia articles. Our approach does not incorporate this specific prior knowledge, as it does not generalize.
>
> We will clarify these differences in the paper to provide a more accurate context for our evaluations.
>
> >"it would be helpful to include standard retrieval metrics (e.g., recall@k or supporting fact accuracy) to better assess the effectiveness of the proposed method in retrieving supporting paragraphs"
>
> We have incorporated retrieval metrics for 9 KILT tasks in Table 7 of Appendix B. For multihop QA tasks, we calculated answer recall@{10, 20, 100} for your reference:
>
> |                       | R@10     | R@20     | R@100    |
> |-----------------------|----------|----------|----------|
> | HotpotQA              | 59.1     | 65.2     | 76.8     |
> | &nbsp;&nbsp; w/ CoRAG | **72.1** | **76.7** | **84.3** |
> | 2WikiMultiHopQA       | 54.9     | 62.1     | 74.6     |
> | &nbsp;&nbsp; w/ CoRAG | **81.4** | **84.8** | **88.8** |
> | Bamboogle             | 31.2     | 40.0     | 57.6     |
> | &nbsp;&nbsp; w/ CoRAG | **59.2** | **68.0** | **75.2** |
> | MuSiQue               | 29.0     | 36.5     | 52.7     |
> | &nbsp;&nbsp; w/ CoRAG | **47.1** | **54.6** | **68.4** |
>
> As demonstrated, CoRAG consistently and significantly improves recall across all datasets, underscoring its effectiveness in retrieving relevant information.
>
> Notes: We followed DPR [1] in calculating recall based on answer matches, as not all datasets provide gold supporting paragraphs. For CoRAG, we utilized reciprocal rank fusion to merge multiple retrieval results into a single ranked list, from which recall was then calculated. For baselines, we use $E5_{\text{large}}$ to retrieve the top 100 passages.
>
> [1] Karpukhin, V., Oğuz, B., Min, S., Lewis, P., Wu, L.Y., Edunov, S., Chen, D., & Yih, W. (2020). Dense Passage Retrieval for Open-Domain Question Answering. ArXiv, abs/2004.04906.
>
> >"Does the proposed framework improve performance when applied to other LLMs beyond LLaMA-3?"
>
> Our proposed method is designed to be model-agnostic and can be seamlessly applied to various LLMs beyond LLaMA-3. For instance, we have successfully applied CoRAG to the Qwen3 family of models, observing similar performance improvements:
>
> | Model                                 | 2WikiQA |      | HotpotQA |      | Bamboogle |      | MuSiQue |      |
> |---------------------------------------|:-----------:|:----:|:------------:|:----:|:-------------:|:----:|:-----------:|:----:|
> |                                       |     EM      |  F1  |      EM      |  F1  |      EM       |  F1  |     EM      |  F1  |
> | Fine-tuned Qwen3-4B w/ $E5_{\text{large}}$ |    49.3     | 55.3 |     45.0     | 57.9 |     32.8      | 43.1 |    13.4     | 23.8 |
> | CoRAG-Qwen3-4B (L=6, greedy)          |  **69.3**   | **74.1** |     **51.6**     | **64.2** |     **49.6**      | **62.5** |    **24.0**     | **34.5** |
> |                                       |             |      |              |      |               |      |             |      |
> | Fine-tuned Qwen3-8B w/ $E5_{\text{large}}$ |    52.1     | 57.9 |     47.1     | 60.0 |     33.6      | 47.6 |    15.3     | 26.3 |
> | CoRAG-Qwen3-8B (L=6, greedy)          |    **70.0**     | **74.8** |     **52.8**     | **66.0** |     **49.6**      | **63.7** |    **25.2**     | **35.9** |
>
> >"The idea of multi-step, dynamic retrieval has been extensively studied in prior work, such as IRCoT, Self-RAG, ITER-RETGEN, and others."
>
> We acknowledge that multi-step retrieval has been explored in prior works as you mentioned. However, to our knowledge, none of the existing literature has investigated the test-time scaling of RAG within a fine-tuning setting. We believe this is a crucial aspect for emerging applications like Deep Research.
>
> An apt analogy would be the distinction between zero-shot CoT prompting and o1-like training for long CoT: while LLMs can generate CoT in a zero-shot manner, explicitly training LLMs to produce longer and higher-quality CoT traces still yields significant benefits.
>
> >"Is there any human evaluation to verify the validity of generated sub-questions and sub-answers?"
>
> Our primary evaluation relies on automatic task metrics. However, for deeper analysis, we also conduct manual inspections of generated sub-questions and sub-answers. Representative examples are provided in Table 9 of the Appendix and in the supplementary material.
>
> >"In the ablation study on retrievers (Line 234), performance gains are attributed to stronger retrievers. Can the authors isolate the contribution of the CoRAG mechanism from that of the retrieval encoder?"
>
> **The ablation results concerning retrievers illustrate that stronger retrievers contribute to improved performance as an orthogonal factor, rather than being the primary source of CoRAG's gains.** In our main results (Table 1), we maintained a consistent retriever (E5$_{\text{large}}$) across various decoding strategies and baselines, including SelfRAG and fine-tuned LLaMA-8B, to isolate and demonstrate the specific contributions of the CoRAG mechanism itself.

---

> ### Author Response · Authors · 2025-08-07
> **Reminder from Authors**
>
> Dear Reviewer gW6e, thank you for your initial review.
>
> We have submitted a rebuttal addressing your concerns, including the clarification of the leaderboard numbers you mentioned and additional results on recall metrics and generalization to other model families. We would greatly appreciate it if you could take a moment to review our response.
>
> Thank you again for your time and consideration.

---

### Official Review · Reviewer_ovoL · 2025-07-07

**Clarity:** 4
**Significance:** 3
**Originality:** 3
**Rating:** 5
**Confidence:** 4

**Summary:**

The paper proposes CoRAG (Chain-of-Retrieval Augmented Generation), a framework that teaches language models to iteratively retrieve, reason over, and reformulate queries before producing an answer. Instead of the single pre-retrieval step used in standard RAG systems, CoRAG creates multi-step “retrieval chains”; these chains are constructed with rejection sampling using an instruction-tuned Llama-3.1-8B model, turning ordinary QA datasets into training data that include intermediate sub-queries, retrieved passages, and sub-answers. A single model is then fine-tuned on three intertwined tasks—next sub-query prediction, sub-answer generation, and final answer generation—so it can decide, at inference time, how many retrieval steps to take and which decoding strategy (greedy, best-of-N, or limited tree search) best balances accuracy against token budget. Experiments on four multi-hop QA datasets show double-digit EM gains over strong baselines, and on the full KILT benchmark an CoRAG-8B model sets new state-of-the-art scores on nearly every task. The authors also conducted comprehensive ablations, demonstrating the effectiveness of each design choice.

**Questions:**

- Not sure if I missed this but what model is used during the rejection sampling stage?

**Ethical Concerns:**

["NO or VERY MINOR ethics concerns only"]

**Final Justification:**

- It's a technically solid paper and the authors have addressed my comments in their rebuttal. I keep my score unchanged.

**Limitations:**

yes

**Quality:**

4

**Strengths And Weaknesses:**

### Strengths
- The proposed training framework -- CoRAG, is technically sound, intuitive, simple, yet quite effective.
- Strong results on a set of multihop reasoning datasets and the CoRAG-8B model achieves a new state-of-the-art performance across all tasks, with the exception of FEVER on KILT benchmark.
- Comprehensive analysis and ablations, clearly showing the how each design choice is made in the framework, which brings insights to the research community.
- The calculation the conditional log-likelihood of “No relevant information found” at test time as a penalty score for each chain is smart and works well.

### Weaknesses
- From my perspective, I don't see major weaknesses of this paper. Here are some of my thoughts that could make the paper more solid: 1) The model is simultaneously trained on three tasks. It would bring more insights to the audience to conduct ablations over the training objective. 2) I would also like to see more experiments regarding the rejection sampling stage, e.g., how different models choices affect the final performance? How about other design choices compared to rejection sampling such as distilling from larger models or using larger models as a judge?

---

> ### Author Rebuttal · Authors · 2025-07-31
>
> Thank you for your encouraging feedback!
>
> >"what model is used during the rejection sampling stage?"
>
> We use the off-the-shelf instruct model (e.g., Llama-3.1-8B-Instruct) for rejection sampling.
> For iterative training, we use the model trained in the previous iteration.
> This approach allows the model to generate its own training data in a bootstrapping manner.
>
> >"It would bring more insights to the audience to conduct ablations over the training objective."
>
> Thanks for the suggestion! One factor we studied is the data sampling ratio for $L_{\text{subquery}}$ and $L_{\text{subanswer}}$. We found that a sampling ratio of 0.2 works well and accelerates training. Increasing the sampling ratio to 1.0 resulted in only a negligible performance drop, as shown in the table below. Both models were evaluated with greedy decoding and a maximum of 6 retrieval steps. We will include this ablation in the revised paper.
>
> | **Model**             | **2WikiQA** |      | **HotpotQA** |      | **Bamboogle** |      | **MuSiQue** |      |
> |-----------------------|:-----------:|:----:|:------------:|:----:|:-------------:|:----:|:-----------:|:----:|
> |                       |     EM      |  F1  |      EM      |  F1  |      EM       |  F1  |     EM      |  F1  |
> | CoRAG-8B (default)    |    70.6     | 75.5 |     54.4     | 67.5 |     48.0      | 63.5 |    27.7     | 38.5 |
> | w/ sampling ratio 1.0 |    69.9     | 74.8 |     53.9     | 67.0 |     48.8      | 63.2 |    26.0     | 36.5 |
>
> > "I would also like to see more experiments regarding the rejection sampling stage, e.g., how different models choices affect the final performance? How about other design choices compared to rejection sampling such as distilling from larger models or using larger models as a judge?"
>
> In Table 3, we present an ablation study comparing the use of smaller LLMs (Llama-3.2-1B and Llama-3.2-3B) with GPT-4o for rejection sampling. The results align with expectations: larger models consistently lead to better performance. Using larger models as a judge is an interesting idea that we will explore in future work.

---

> > ### Comment · Reviewer_ovoL · 2025-08-06
> > **Thanks for the reply**
> >
> > Thanks the authors for the response. I will keep my score unchanged.

---

### Note · Authors · 2025-08-13

We would like to express our sincere thanks to all reviewers for their insightful feedback, which has been instrumental in strengthening our paper's clarity and rigor.

Our work presents CoRAG, a simple yet powerful training framework designed for test-time scaling of RAG. We've demonstrated state-of-the-art performance across multiple open-domain QA benchmarks and conducted a systematic, comprehensive study on test-time scaling for RAG in a fine-tuning setting. Our approach is model-agnostic and robust, as shown by consistent gains across different LLM backbones.

In response to the reviewers' comments, the final manuscript will be updated to:
* Clarify our task settings and leaderboard comparisons to avoid any possible misinterpretation.
* Provide a more detailed analysis of retrieval performance, including a broader range of metrics.
* Incorporate a discussion of related R1-Zero-like RAG methods and relevant reinforcement learning literature.
* Deepen our understanding of failure modes and the effectiveness of generated sub-questions.

We note that Reviewer gW6e did not participate in the discussion phase. We are eager to know whether our rebuttal has fully addressed their concerns, especially regarding leaderboard comparisons and ablation interpretations.

Thank you once more for your valuable time and feedback.

---

### Decision · Program_Chairs · 2025-09-17

**Decision:**

Accept (poster)

**Comment:**

This paper introduces CoRAG (Chain-of-Retrieval Augmented Generation). CoRAG is a  framework for training RAG systems to perform multi-step retrieval to answer complex queries.

The reviewers noted the following strengths of the paper:
* The paper is well-written and the proposed method clearly explained
* The proposed framework is novel and technically sound
* CoRAG achieves strong empirical results, establishing a new SOTA on the KILT benchmark as well as showing significant gains on multi-hop QA tasks
* The paper presents a comprehensive analysis of test-time compute scaling that demonstrates a trade-off between performance and token consumption

The reviewers also identified a number of weakness with the paper, including:
* There were some initial concerns about unfair experimental conditions with respect to few-shot baselines and leaderboard scores
* The novelty of the approach was also questioned, as the high-level concept of multi-step retrieval was previously explored

The authors provided convincing rebuttal that adequately addressed the primary concerns raised by the reviewers. The rebuttal clarified that the leaderboard comparisons were indeed not applicable due to differences in task settings. It highlighted the paper does include a direct and hence more fair comparison to a fine-tuned baseline, which CoRAG outperforms. Furthermore, the rebuttal included additional experimental evaluations focused on retrieval recall and model generalization, helping to strengthen the paper. The consensus among the reviewers is positive, as the initial weaknesses raised were largely resolved.

Overall, this is a high-quality paper with strong results. Therefore, the final decision is to accept this paper.